# Image-Guided Localization Techniques for Metastatic Axillary Lymph Nodes in Breast Cancer; What Radiologists Should Know

**DOI:** 10.3390/cancers15072130

**Published:** 2023-04-03

**Authors:** Valerio Di Paola, Giorgio Mazzotta, Marco Conti, Simone Palma, Federico Orsini, Laura Mola, Francesca Ferrara, Valentina Longo, Enida Bufi, Anna D’Angelo, Camilla Panico, Paola Clauser, Paolo Belli, Riccardo Manfredi

**Affiliations:** 1Department of Bioimaging, Radiation Oncology and Hematology, UOC of Radiodiagnostica Presidio Columbus, Fondazione Policlinico Universitario A. Gemelli IRCSS, Largo A. Gemelli 8, 00168 Rome, Italy; 2Institute of Radiology, Catholic University of the Sacred Heart, Largo A. Gemelli 8, 00168 Rome, Italy; 3Department of Bioimaging, Radiation Oncology and Hematology, UOC of Radiologia Toracica e Cardiovascolare, Fondazione Policlinico Universitario A. Gemelli IRCSS, Largo A. Gemelli 8, 00168 Rome, Italy; 4Department of Biomedical Imaging and Image-Guided Radiotherapy, Division of Molecular and Gender Imaging, Medical University of Vienna, Waehringer Guertel 18-20, 1090 Vienna, Austria

**Keywords:** breast cancer, axillary lymph node, sentinel lymph node biopsy, axillary lymph node dissection, targeted axillary dissection, NACT, localization techniques, guidelines

## Abstract

**Simple Summary:**

Breast cancer is the most frequent cancer affecting women, and axillary lymph nodes (ALNs) are the most common initial site of metastatic spread. In patients with positive ALNs undergoing neoadjuvant chemotherapy (NACT), it is necessary to localize and identify the lymph node metastases in order to perform less invasive axillary surgery, such as targeted axillary dissection (TAD). In this setting, the choice of the most appropriate localization methods is crucial to correctly orientate the removal of the pathological ALNs. This is more important considering that ALNs can become non-palpable after NACT. National Comprehensive Cancer Network (NCCN) guidelines also suggest their possible use in a non-NACT setting, particularly in patients candidate to SLNB with limited numbers of positive ALNs in whom ALNs have been biopsied.

**Abstract:**

Targeted axillary dissection (TAD) is an axillary staging technique after NACT that involves the removal of biopsy-proven metastatic lymph nodes in addition to sentinel lymph node biopsy (SLNB). This technique avoids the morbidity of traditional axillary lymph node dissection and has shown a lower false-negative rate than SLNB alone. Therefore, marking positive axillary lymph nodes before NACT is critical in order to locate and remove them in the subsequent surgery. Current localization methods include clip placement with intraoperative ultrasound, carbon-suspension liquids, localization wires, radioactive tracer-based localizers, magnetic seeds, radar reflectors, and radiofrequency identification devices. The aim of this paper is to illustrate the management of axillary lymph nodes based on current guidelines and explain the features of axillary lymph node markers, with relative advantages and disadvantages.

## 1. Introduction

Breast cancer is the most common cancer in females [1,2]. Axillary lymph nodes (ALNs) represent the most common initial site of metastatic spread, and lymph nodes (LNs) status is the major prognostic factor [3].

Historically, axillary staging has been surgically assessed by axillary lymph node dissection (ALND). 

This procedure is burdened by a high variable percentage rate of arm lymphedema (7–77%) due to the interruption of the corresponding axillary lymphatics and/or removal of the LNs draining the upper limb.

Therefore, several axillary LNs classifications have been made to orientate surgeons during axillary surgery. The two main methods are Clough’s classification and Li’s classification [4]. Clough’s classification is based on the intersection of two anatomical structures, the second intercostobrachial nerve (ICBN) and the lateral thoracic vein (LTV): zone A is the area that extends from the lower margin of the axilla to the second ICBN; zone B is the area that extends from the second ICBN to the axillary vein under the pectoralis minor muscle; zone C is the lateral axillary area external to zone A; and zone D is the lateral axillary area external to zone B. The study documented that in around 98% of patients the axillary sentinel lymph nodes (SNLs) were located medially to the LTV, either in zone A below the second ICBN or in zone B above it. According to these findings, surgeons should avoid unnecessary more lateral axillary dissections [5]. 

Li’s classification divides the axillary space by the ICBN into an upper part (A) and a lower part (B). In this series, all the SLNs were found under the ICBNs. Moreover, the presence of metastases in part A was always associated with metastases in part B; on the contrary, if no metastasis was present in zone B, then the LNs of part A were also not metastatic. In this light, the authors recommended that ICBN might be considered potential candidates as an anatomic landmark during the procedures of ALND [6]. The above classifications may help to avoid lymphedema caused by unnecessary dissection of the axilla in the area lateral to the LTV and upper the ICBN.

However, questions remain regarding whether the preservation of these LNs affects oncological risk.

ALNs management in breast cancer patients has been rapidly changing in the past two decades, favoring less invasive approaches. Given its high morbidity, ALND has been gradually replaced in some conditions by sentinel lymph node biopsy (SLNB), a minimally invasive method based on surgical excision and pathological analysis of the first draining LN(s) with a direct lymphatic connection to the primary tumor site [7]. The most widely used techniques for sentinel lymph node (SLN) identification are based on the use of a radioactive tracer (Technetium-99), blue dye, or both (“dual tracer”). The radioactive tracer-based method involves intra- and subdermal injection at the skin margins overlying the tumor of Tc-99m-labeled nanocolloidal albumin before surgery, followed by a preoperative lymphoscintigram obtained hours after tracer administration. A gamma probe identifies radioactivity in the draining ALNs during surgery [8]. If blue dye is used, blue-stained lymphatic channels visualized during surgery are followed to LNs where the blue dye accumulates [9]. SLNs are resected and submitted for pathological analysis.

New tracers, such as near-infrared (NIR) imaging agents, including indocyanine green (ICG), have demonstrated a similar accuracy to the radioactive tracer (94.3% vs. 96.2%), with the advantage of higher availability, lower costs, and lack of radioactivity exposure to patients and surgeons. The accuracy was high also after NACT, with 95.4% for each tracer [10].

The efficacy of another technique after NACT, such as superparamagnetic iron oxide (SPIO), was recently investigated. It consists of a magnetic tracer originally developed for contrast-enhanced magnetic resonance imaging. It flows through the lymphatic system and is trapped by the SLN, becoming detectable by a hand-held magnetometer. The SPIO method showed a significantly higher chance versus the radioactive tracer method in both retrieving (71% vs. 11.3%; OR = 19.21; *p* < 0.0001) and evaluating (71.4% vs. 51.6%; OR = 3.21; *p* = 0.0032) at least three SLNs [11]. 

SLNB is the standard of care for axillary staging in early T1–T2, clinically node-negative breast cancer [12]. National Comprehensive Cancer Network (NCCN) guidelines have expanded the indications for SLNB, also suggesting the procedure for patients with T1 or T2 stage, clinically node-negative and from 0 to 2 suspicious ALNs at imaging or metastatic LN confirmed by needle biopsy [13].

Given the great success of SLNB and the potential complications related to ALND, this technique has emerged as a surgical staging strategy even in patients treated with NACT (NACT).

Even if SLNB has shown excellent accuracy in patients who are lymph node-negative on clinical and ultrasound examination before NACT [14,15], becoming the standard of care in this category of patients, the management of patients with metastatic LNs at diagnosis undergoing NACT has been challenging (Figure 1). Modern NACT can lead to a pathologic complete response (pCR) of ALN metastasis in approximately 40–75% of patients [9,16,17,18].

Four prospective, multicenter trials (SENTINA, ACOSOG Z1071, SN FNAC, and GANEA 2) have evaluated SLNB in this category of patients. In these studies, patients underwent SLNB followed by ALND; the primary endpoint was to determine the false-negative rate (FNR), of which a threshold of 10% was arbitrarily chosen but widely accepted.

The false-negative rate obtained was unacceptable and higher than the established cut-off, with FNRs of 14.2% [19], 12.6% [9], 14.2% [20], and 11.9% [15], respectively.

Chemotherapy response may alter axillary drainage due to fibrotic reaction [21], and this may explain the false-negative rate.

To reduce the FNR of SLNB, different approaches have been proposed: the use of immunohistochemistry to detect isolated tumor cells and micro-metastases [20], the dual mapping technique [9,22,23], and resections of more than one SLN (Table 1).

Resection of three SLN showed low FNR, but only a limited number of patients (SENTINA, 34%; ACOSOG Z1071, 56.3%) had three or more SLN removed.

However, metastatic LNs at diagnosis do not always coincide with SLNs after NACT [24,25,26].

Therefore, marking positive LNs at the time of diagnosis allows their removal during post-NACT surgery. Targeted axillary dissection (TAD) is an axillary staging technique that combines the removal of the metastatic LNs clipped before NACT and SLNB. This technique has recently been introduced in the main guidelines (Table 2) and is associated with a further reduction in the false-negative rate, which ranges from 2% to 6.8% [24,25,27,28,29,30,31,32,33].

In this scenario, the placement of ALN markers is crucial for the management of the patients with positive LNs at diagnosis who show the complete axillary response after NACT. NCCN guidelines suggest their possible use in a non-NACT setting, particularly in patients candidate for SLNB with limited numbers of positive ALNs in whom ALNs have been biopsied. In these cases, if a positive LN is marked at biopsy, every effort should be made to remove the marked LN(s) at the time of surgery [13].

The current localization methods include clip placement with intraoperative ultrasound, carbon-suspension liquids, localization wires, radioactive seeds, magnetic seeds, radar reflectors, and radiofrequency identification devices. Some markers can be localized with preoperative ultrasound; their identification during surgery can be facilitated by intraoperative ultrasound or, in the case of a radioactive seed or a magnetic clip, with the corresponding detector probe.

The features, advantages, and disadvantages of each method are presented below.

## 2. Axillary Localization Techniques

### 2.1. Marker Clips and Intraoperative Ultrasonography

Marker clips are the most widespread method used for localizing LNs in the axilla [38]. They are typically placed during the needle biopsy procedure, which can be ultrasound-guided [39], stereotactic [40], or MRI (Magnetic Resonance Imaging)-guided [41], using a 9–18-gauge device with a sliding or plunger system.

Although many types of clips have been described, two main categories can be distinguished: traditional metallic clips, usually made of titanium or steel (Figure 2), and metallic clips that are centrally embedded in a hygroscopic bioresorbable plug (typically polymers such as collagen but also polyglycolic or polylactic acid), which tend to absorb water over time [42], expanding their volume. Water absorption makes the device sonographically visible so that future localizations can be performed using ultrasound. In fact, these devices are more sonographically visible during the first weeks after their placement; then, their visibility has been demonstrated to gradually decrease in the long term up to 12–15 months, at which time they usually cannot be identified anymore [43].

One of the greatest advantages of sonographically visible marker clips is the possibility to use them in conjunction with intraoperative ultrasound, which makes it possible to avoid to perform a preoperative localization procedure since the clip is deployed at the time of biopsy [44]. This technique offers high rates of clear surgical margins in breast lesions, which are approximately 90–96.2% [45,46]. On the other hand, its success depends on the visibility of the clip which tends to decrease over time, as mentioned above, requiring good familiarity with ultrasound [47].

Another advantage is that biopsy marker clips do not involve radioactivity, so they can be retained in the body for an indefinite time, unlike radioactive seeds that require a strict timeframe for their removal.

Collagen-based biopsy clips have the benefit of decreasing bleeding and hematoma formation due to the hemostatic effect of collagen [43].

Non-hygroscopic clips have the advantage of very small measurements (about 3 mm); therefore, even if their small size reduces their visibility, they can be deployed directly in the cortex of a LN or in adjacency to the biopsied LN [48].

The costs of both the traditional metallic clip and the ultrasonographically visible hygroscopic clip are similar and relatively low.

The important disadvantages of the marker clip are clip migration and clip extrusion. This displacement typically occurs in the *z*-axis and is called the “accordion effect,” when the breast is compressed during the biopsy procedure and the clip is deployed in the adjacent tissue rather than in the LN. As the compression is released, the clip tends to move away from its original site. Even if the occurrence of this phenomenon is not predictable, it is described to occur in around 50% of the cases in the breast and may be even more frequent in the axilla due to the arm movements [48]. However, the clip shift is usually <1 cm, which is considered acceptable.

Moreover, some studies suggest that it is more frequent when using hygroscopic ultrasonographically visible clips because, as they increase in size, they are more likely to move along the biopsy track [49]. Obviously, the farther a clip is located from the actual biopsy site, the less it will be useful as a target for future localizations, resulting in a relatively low detection rate.

Moreover, a clip may sometimes lead to a reaction of the node tissue, which can be misinterpreted on pathological examination. Rare cases of allergic reactions have been described, mostly related to the presence of nickel in some titanium clips [43].

### 2.2. Carbon Suspension-Based Localization

Tattooing with carbon suspension has been widely used for marking lesions or tumors biopsied during colonoscopy. This technique has also been used to mark metastatic ALN before NACT, although the use of carbon solution is off label for this setting, being licensed only for the gastrointestinal tract and primitive breast lesions.

The procedure is performed at the time of LN biopsy by injecting a charcoal suspension into the cortical of the suspected LN under ultrasound guidance. Immediately after injection, the charcoal particles are visible with a hyperechoic halo around the LN [50]. The black-marked perinodal tissues are visually localized intraoperatively.

Carbon suspension-based localization of metastatic LN is a simple, radiation-free, low-cost technique that is well tolerated by the patient. Tattooing is performed at the time of biopsy and remains for up to 6–8 months after injection [43], so it has the advantage of not requiring an additional localization procedure before surgery.

Side effects are rarely described. Only one study reported the development of foreign body granulomas in 3% of patients [51]. Data obtained from the multicenter, prospective TATTOO study of 110 patients revealed that the only side effect associated with charcoal injection was permanent tattooing of the axillary skin, which occurred in five patients (4.5%). In four of these, the injection channel from the marked LN to the skin was tattooed, so the incidence of this side effect could potentially be reduced using the correct injection technique [52]. Although limited data are available, a detection rate between 94 and 100% has been reported [53,54,55]. The FNR for TAD by carbon suspension-based localization reported by the TATTOO study is 9.1%; this is below the established cut-off of 10% but higher than previous studies of TAD. The authors explained this by a different definition of SLN and by the smaller number of patients who were node-positive after primary systemic therapy [52].

However, this procedure has some disadvantages. Tattooed LNs are visually localized, so surgical exploration of the axilla is required, which is more invasive than probe-assisted techniques.

The presence of skin tattoos in the upper body can make more difficult the identification of marked metastatic LNs. In these patients, there may be other pigmented ALN resulting from the drainage of ink from skin tattoos. The distinction may be allowed by the fact that usually perinodal tissue is pigmented only in iatrogenic LN tattooing [50].

It is important to keep in mind that black pigment may migrate from one LN to another. This has a practical repercussion, as all tattooed LNs must be found macroscopically during surgery. Macroscopically non-black LNs with microscopically detected small foci of carbon particles should not be considered as retrieved marked LNs [56].

Choy et al. report that in three out of twenty-eight cases, black tattooed LNswere identified during surgery, but the ink was not found histologically. This could be due to the accidental removal of the marked adipose tissue during specimen processing [54].

Conversely, if the injection of the charcoal solution occurs deeper within the node and/or too small volumes of solutions are used, then the marked LNs may not appear black macroscopically but contain pigment only microscopically [54]. This could lead the surgeon to extend the dissection in an attempt to locate the marked LNs, making the procedure more invasive.

Another problem might arise if a SLNB by blue dye is performed at the same time. The appearance of the tattooed LNs could mimic that of the blue dye used for the SLN and make its identification more difficult. The tattoo ink is grayish-black, while the isosulfan blue is an intense blue, but sometimes the distinction might be subtle [54]. In addition, the amount of carbon solution injected varies from 0.1 to 0.5 mL. The volume to be injected could be modulated according to the size of the LNs, but the optimal injection volume remains unclear [54].

### 2.3. Metal Wires

The use of thin metal wires for localizing non-palpable lesions is a well-established technique in the breast, and it could be taken into consideration for axillary LNs.

Localization wires are usually deployed encased in a 19-gauge hollow needle and placed in the targeted abnormality with the guidance of ultrasonography, mammography, MRI, and, more rarely, CT (Computed Tomography) [57] (Figure 3, Figure 4 and Figure 5).

Wires are typically 20-gauge sized, made of surgical steel, and present an anchoring tip that fixes them to the targeted tissues. They can specifically be distinguished into two main categories, fixed and repositionable. Unlike fixed wires, which cannot be displaced once they are deployed, the anchoring tip of repositionable wires has a configuration that allows them to be retracted back into the hollow needle in order to reposition the wire if necessary [58].

Once positioned, the wire protrudes from the skin and can be easily visualized by the surgeon, guiding him to the targeted LN. This technique offers good rates of clear surgical margins, which are approximately 70–87% in breast lesions [59], with the advantage of very low costs.

Nevertheless, wire localization systems are infrequently used in the axilla by radiologists and surgeons because of some disadvantages.

First, this technology inevitably needs to couple the radiologic and surgical workflow, as wire placement and its removal need to be performed on the same day; consequently, this may lead to some scheduling problems and sometimes delays in the surgery workflow [60].

Another disadvantage consists in the fact that metal wires are often associated with complications such as pain, bleeding, hematomas, and injuries of the surrounding soft tissues, which can be particularly concerning in the axilla, where many critical structures are located, such as the brachial plexus and the axillary artery and vein [59].

Another complication of wire systems is migration, which has been described quite infrequently in the breast [61,62] and can be of great distance in some cases. The risk of migration could be higher in the axilla rather than the breast, as this region is more subject to movements because of the shoulder girdle contractions [63,64].

Moreover, according to the literature, another infrequent risk connected to metal wires localization in the breast is wire transection. The removal of eventual wire fragments could be particularly challenging in the axilla rather than in the breast due to the higher anatomical complexity of this area [65].

### 2.4. Magnetic Seed

Magnetic seed localization is a relatively recent technique, currently used in the breast, which could be adapted for localizing axillary LNs.

Magnetic seeds consist of millimetric (1–5 mm) paramagnetic low-nickel steel pellets which are positioned using an 18-gauge steel needle under ultrasonographic and or mammographic guidance (Figure 6). The seed is then localized by the surgeon with a probe which, by generating an alternating magnetic field, magnetizes the seed and makes it recognizable in the tissue. The probe both shows a numerical count and produces an audio tone in order to best indicate the location of the marker [66]. The operator deploys the seed by pushing on an obturator, similarly to the placement of biopsy marker clips [67]. According to the literature, magnetic seeds can be placed at depths up to 3–4 cm and, even if they were initially approved to be retained in the body for a maximum of 30 days, their use has been recently extended for a longer term [68].

Recent studies regarding the use of this technology in the breast suggest that the seeds do not tend to migrate and can be detected at a maximum depth of 3.5 cm [69], even if deeper lesions can be also localizable with the combined use of the “intraoperative palpation,” which consists in using downward pressure on the probe during the localization.

Adapting the use of magnetic seeds localization in the axilla could have several advantages. First, this technology allows decoupling the radiological and surgical workflows, with the benefit that the seed does not involve radioactivity and can be retained at least up to 30 days before the surgery, without any decay in signal over time [69]. Moreover, the seed is deployed at the tip of the needle, which theoretically could facilitate placing it inside a LN with a low risk of damaging the axillary critical structures.

An important disadvantage of this technique is depth limitation. Even if 3.5 cm can be a sufficient depth for an adequate localization in the axilla in most cases, women with high body mass index or LNs located deep in the axilla may require a higher depth to be localized; this could partially be compensated by the use of the “intraoperative palpation” technique [58].

Another drawback is represented by the relatively high costs, even if they could be partially mitigated by potential improvements in workflow and volume discounts.

Other limitations are related to the use of a magnetic-based technology, which involves susceptibility to other magnetic signals, requires calibration before and during surgery, and requires the use of non-magnetic polymer tools by the surgeon [70].

Finally, if the magnetic seed is placed before the MRI, it may determine the formation of susceptibility artifacts, with consequent difficulty to examine axillary details [69] (Figure 7).

### 2.5. Radar and Infrared Light

The use of the micro impulse radar and infrared light is a non-radioactive and non-wire-based technique currently used for the breast, potentially adaptable in the axilla.

This technology is based on placing a small reflector clip flanked by two nitinol antennae; both the reflector and the antennae have a length of 4 mm, with a total length of 12 mm. The device is placed under ultrasound or mammographic guidance, using a 12-gauge steel needle, in which the reflector is located at the tip. The radar reflector is localized by activating it with an infrared light source produced by a handpiece connected to a console; the handpiece and the console detect the reflected signal, producing an audible tone and showing a numerical indicator, which provides information about the location of the device [71]. The reflector has recently been approved to be retained in the body for an indefinite time [70].

According to the literature, the handpiece to reflector accuracy is around 1 mm, and this system makes it possible to perform breast localizations up to a maximum depth of 6–8 cm, with the highest rates of surgical success around a depth of 4.5 cm [72]. Only one study has reported the case of a reflector migration due to the presence of a hematoma [73]; another study from the same author reported two cases of non-detection of the reflector, respectively, secondary to a hematoma and a large calcified fibroadenoma [74].

A study aimed to evaluate the efficacy of radar and infrared light localization in the axilla demonstrated that, among 19 patients with biopsy-proven LN metastases, this technology allowed the successful localization of all the pathological nodes, which were all consequently excised with success [75].

The potential use of radar and infrared localization in the axilla has several advantages. Similarly to other techniques, this technology allows for the decoupling of the radiological and surgical workflows and may be able to facilitate the surgical planning in the axilla, as it has proven to be very accurate [76].

Moreover, since there are no restrictions on the length of time the reflector can be retained in the body, it could also be placed before the NACT, even with the drawback of susceptibility artifacts when an MRI is performed.

A possible disadvantage is that cautery and the presence of halogen operating room lights can affect the retrieval of the reflector clip [72]. Specifically, the reflector clip has a relatively large size (12 mm) in relation to the average size of an axillary LN so that, in some cases, it could be challenging to directly deploy it in the cortex of a LN, and it may be necessary to place it its adjacency, resulting in some confusion regarding which node was previously biopsied and marked, especially if other LNs are located nearby. Moreover, the presence of structure as muscles and vessels which are normally located in the axilla may impede the signal from the detector, potentially impacting the chances of localizing LNs [58].

Another drawback is the possibility of allergic reactions to nickel, which means that the reflector cannot be placed in patients who are allergic to this metal [43].

Finally, this technology has relatively high costs, even if they could be partially mitigated by potential improvements in workflow and volume discounts.

### 2.6. Radioactive Tracer-Based Localization

#### 2.6.1. Radioactive Seed Localization

Since 2008, Radioactive Seed Localization (RSL) has been used to localize not only breast lesions but also small and non-palpable metastatic LNs in the neoadjuvant setting. The RSL consists of a titanium seed, which measures about 4 mm × 0.8 mm, which contains a core made of aluminum and copper-coated gold, covered in radioactive Iodine-125 [58]. The seeds emit photon radiation, and those used in ALNs nodes have an activity between 1.6 and 70 MBq with a half-life of 60 days and a 27 keV gamma radiation emission peak [77].

The seeds are kept at the radionuclide laboratory within the department of Nuclear Medicine and are used under the control of an authorized staff that ensures the seeds are transported in a sterile container or pre-loaded into an 18-gauge spinal needle, which is blocked with specific materials in order to avoid premature deployment [58,77]. The maximum amount of time for seeds permanence recommended by guidelines is between 5 and 7 days [58]. The seed is implanted using ultrasound radioguidance. Subsequently, surgery is guided by audible and visual feedback from a probe [78].

Several studies have demonstrated the high efficacy of removing pathological LNs marked through RSL (100%) even after neoadjuvant therapy [58]. In a randomized clinical trial by Bloomquist et al., RSL-arm demonstrated a better outcome in terms of intra-procedure pain compared to wire guide localization. The overall suitability of the procedure was evaluated as very good to excellent in 85% of RSL patients compared with 44% of wire guide localization patients [79].

RSL has better patient compliance compared to wire guide localization. In fact, transcutaneous localization before skin incision is possible [78].

The equipment to perform LNs localization, especially the probe and console, are burdened by relatively high costs. However, the costs can be amortized by use of the same equipment for the SLN research technique and other radio-guided surgery procedures, not exclusively for the breast.

Despite the small dose of radiations and low radioactive risk for the operators [58], the procedure is not authorized in some countries [79] and requires complex radiation safety procedure and a strict timeframe for removal. One study showed that dropping of semen into the surgical bed can occur during LN removal, even if with a very low possibility. This happens when the semen is placed at the edge of the LN [58]. An additional disadvantage related to the misplaced semen is the impossibility of repositioning the seed [79]. In these cases, the only resolving alternative is surgical removal.

Moreover, a signal reduction over time has been described in some cases of prolonged or interrupted chemotherapy. Despite a very low risk, emergency treatment with iodine is required in case of semen rupture to avoid damage to the thyroid gland [79].

However, the risk is very low due to the negligible amount of radioactivity contained in the seed.

One study showed that the RSL/SLNB combination is a promising approach for axillary staging for patients with biopsy-proven axillary metastasis and disease becoming cN0 after NACT. In this regard, axillary staging with SLNB alone has a false-negative rate of more than 10%. This happens because of variations in lymphatic circulation or chemotherapy-related LN changes. On the other hand, axillary status after treatment can be staged more accurately with localization of the LN previously sampled by biopsy in conjunction with an SLNB. Surgical localization with 125I seed is achieved in 97% of cases. The MD Anderson and the Netherlands Cancer Institute demonstrate a false-negative rate of 2.3% when the assessed LN is retrieved in conjunction with SLNB. These are promising data, but they require further evidence in the literature [80].

#### 2.6.2. Radioguided Occult Lesion Localization (ROLL)

Radioguided Occult Lesion Localization (ROLL) is a relatively recent method to localize and orientate the excision of non-palpable breast lesions, inspired by SLNB.

This technique consists in injecting ^99m^Tc-labeled human serum albumin adjacent to the lesion under ultrasonographic or stereotactic guidance within 24 h before surgery (Figure 8).

Subsequently, a surgical biopsy is performed using a hand-held gamma ray detection probe designed for SLN localization, which allows for the identification of the area of maximal radioactivity in order to mark the site of the lesion [81,82].

In clinical practice, the radioisotope injection can be performed either on the same day of surgery [83] or on the day before surgery [84].

Injecting the radioisotope on the same day of surgery requires the schedules of the nuclear medicine, the radiology department, and the operating room to be strictly coordinated, potentially resulting in scheduling problems. On the other hand, a day-before-surgery protocol requires the administration of an up to a two-fold dose of the radioactive tracer, which represents its major drawback.

According to the literature, the 2-day protocol has proven to be the most useful for many reasons. First, ROLL is associated with minimal radiation exposure for both patients and medical staff, mainly because of the low levels of the injected activity and the optimal characteristics of ^99m^Tc (ease of labeling, short half-life of 6 h). As a result, the dose absorbed from the inoculated area is negligible and concentrated within the removed tissue [85].

Moreover, the 2-day protocol allows the patients to undergo the procedure in a calm setting, and they do not need to be nil per oral. It provides an optimal timeframe for nuclear medicine physicians to prepare the radiotracer and send it to the radiology department where the radiologist will inject it into the patient’s breast, and for radiologists to interpret the radiological findings on ultrasound and mammography with more accuracy. The surgery could be scheduled as the first case on the following day [86,87].

As a result, ROLL has been demonstrated to allow a less invasive, relatively fast, and accurate removal of non-palpable breast lesions compared with the conventional localization techniques, reducing the rate of reoperation and resulting in relatively low costs [88]. According to the literature, the rate of unclear surgical margins in the breast is reported to be 11–30% in different series [89,90,91,92].

Due to its several advantages, ROLL technology could be adapted for localizing axillary LNs.

A recent study demonstrated that it is technically feasible to perform ROLL on previously clip-marked LNs and selectively remove them at surgery; specifically, in this study, identifying the ROLL-marked node with clip both preoperatively and at surgery was successful in 87% (33/38) of the procedures [93]. 

Another great advantage of this technique is the possibility to perform both ROLL and SNB within one surgical session, which has led to the development of the SNOLL technique (Sentinel Node and Occult Lesion Localization), which allows the removal of a non-palpable breast lesion together with SLN in a single surgical procedure. This seems to be a feasible method to be taken into consideration in patients treated with breast-conserving therapy [94].

### 2.7. Radiofrequency Identification Devices

Radiofrequency identification devices (RFID) are one of the newest options for ALNs localization. The technique consists of the insertion of a little coil and a microchip, which are localized in a glass casing that measures about 12 mm × 2 mm. A 2 mm skin incision is necessary to allow the position of the applicator; then, the tag is located through a hollow needle [58,95]. This step is critical as the use of the applicator alone could damage axillary structures. For this reason, continuous ultrasound surveillance is recommended [95].

The tag can be identified by a probe, which sends out a radiofrequency signal which is captured by the tag and then sent back to the probe. Subsequently, through a sound signal, a device shows how far the probe is from the tag [58]. Women with a lesion located deeper than 7 cm from the skin when lying supine cannot undergo RFID localization due to the maximum reach of the RFID reader of 7 cm [96]. According to a published study on 10 patients, no preoperative, intraoperative, or postoperative complications were observed during localization [97].

In cases where more than one microchip is inserted inside the armpit, the microchip identification number allows them to be distinguished [79]. Although this is not an absolute contraindication as it rarely occurs, operators should be careful not to place tags within 20 mm of each other, as the reader may not distinguish between them [98].

The FDA has approved device insertion only for breast lesions up to 30 days before surgery, but it has not yet been approved for the axilla [58].

RFID has many advantages. First, it is highly accurate, with an intraoperative detection rate for RFID-tagged target lymph nodes (TLN) of 100% [95]. In addition, another study showed the superiority of RFID over magnetic seed in identifying the exact distance between the probe and tag so that very precise TAD can be performed. Differently from RSL, radioactivity is not involved, and the signal does not decrease over time in cases of prolonged or interrupted chemotherapy. RFID has a better patient compliance compared to wire guide localization. In fact, transcutaneous localization before skin incision is possible [58]. Some studies have supported better patient compliance and, according to radiologists’ and surgeons’ opinions, it has a better reliability compared to wire guide localization [79].

On the other hand, it also has some disadvantages, such as the lack of trials affirming its reliability and the high costs of the device. First, the costs of RFID tags, including the ultrasound, placement of the clip, RFID applicator, and RFID tag, are approximately twice as expensive as regular titanium clips, without including indirect costs [99]. Moreover, the glass cover could result in the occurrence of artifacts at MRI. There are concerns regarding the use in patients with pacemakers and Automatic Implantable Cardioverter Defibrillators. For this reason, it should be excluded in these patients [79]. In addition, there may be difficulties in placement in the LNsdue to excessive size [58]. Lastly, there is no possibility to reposition the seed in the case it is misplaced [79].

Some have suggested that the size of the needle could be a cause of the displacement of the seed. In addition, insertion of the tag into hard masses, because of its large size, can be severely difficult for operators.

## 3. Conclusions

Historically, axillary staging has been surgically assessed by axillary lymph node dissection (ALND). Given its high morbidity, ALND has been gradually replaced by sentinel lymph node biopsy (SLNB), which is the standard of care for axillary staging in early T1–T2, clinically node-negative breast cancer [12]. NCCN guidelines have expanded the indications for SLNB to patients with T1 or T2 stage with 0–2 suspicious ALNs at imaging or metastatic LNs confirmed by needle biopsy [13].

Moreover, in the NACT setting, marking positive LNs at the time of diagnosis allows for their removal during post-NACT surgery because metastatic LNs at diagnosis do not always coincide with SLNs after NACT [24,25,26]. Furthermore, the evaluation of node involvement must be performed both at the time of diagnosis and after the end of NACT^96^.

Targeted axillary dissection (TAD) is an axillary staging technique that combines the removal of the metastatic LNs clipped before NACT and SLNB.

This technique has recently been introduced in the main guidelines and is associated with a further reduction in the false-negative rate, which ranges from 2% to 6.8% [24,25,27,28,29,30,31,32,33].

In this scenario, the placement of ALN markers is crucial for the management of the patients with positive LNs at diagnosis who show a complete axillary response after NACT. NCCN guidelines suggest their possible use in a non-NACT setting, particularly in patients candidate for SLNB with limited numbers of positive ALNs in whom ALNs have been biopsied. In these cases, if a positive LN is marked at biopsy, every effort should be made to remove the marked LN(s) at the time of surgery [13].

Current localization methods include clip placement with intraoperative ultrasound, carbon-suspension liquids, localization wires, radioactive seeds, radioactive tracer-based localizers, magnetic seeds, radar reflectors, and radiofrequency identification devices. Some markers can be localized with preoperative ultrasound; their identification during surgery can be facilitated by intraoperative ultrasound or, in the case of a radioactive seed or a magnetic clip, with the corresponding detector probe.

The knowledge of different types of localization methods and their relative advantages and disadvantages is crucial to orientate the choice among the most appropriate method, which should also take into account the specific resources of the various reference hospitals. The development of more integrated international guidelines on the choice of individual localization methods and their clinical indications seems appropriate.

## Figures and Tables

**Figure 1 cancers-15-02130-f001:**
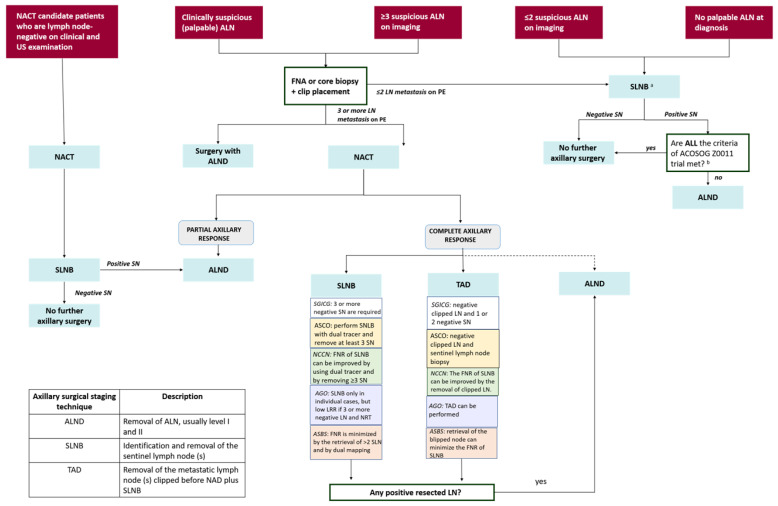
ALN management is illustrated based on current guidelines. ^a^ If a positive LN is clipped at biopsy, every effort should be made to remove the clipped node at the time of surgery. ^b^ They include: T1 or T2 tumor, clinically negative nodes, 1 or 2 positive nodes on SLNB, planned breast-conserving surgery, planned whole-breast radiation therapy, no NACT planned. Abbreviations: FNA: fine needle aspiration; PE: pathological examination; NRT: nodal radiotherapy; LRR: local recurrence rate; NACT: Neoadjuvant chemotherapy; SLNB: sentinel lymph node biopsy; TAD: targeted axillary dissection; ALND: ALN dissection; FNR: false negative rate; SGICG: St. Gallen International Consensus Guidelines; ASCO: American Society of Clinical Oncology; NCCN: National Comprehensive Cancer Network; AGO: Breast Committee of the German Gynaecological Oncology Working Group; ASBS: American Society of Breast Surgeon.

**Figure 2 cancers-15-02130-f002:**
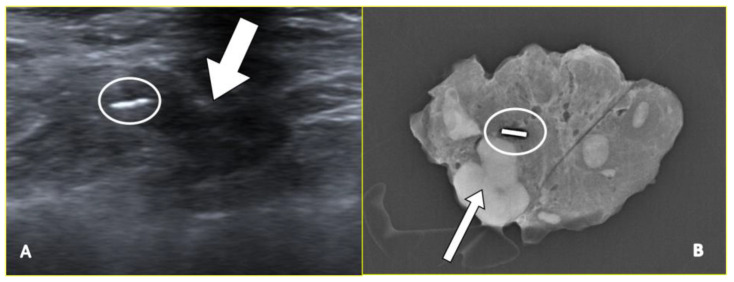
A 65-year-old patient with left breast cancer and ipsilateral left axillary node metastasis treated with neoadjuvant chemotherapy. The ultrasound image (**A**) shows a hyperechoic marker clip (circle) in the hypoechoic pathological axillary node (arrow). Subsequently, the post-surgical specimen radiogram (**B**) confirmed the presence of the markers (circle) adjacent to the pathological node (arrow).

**Figure 3 cancers-15-02130-f003:**
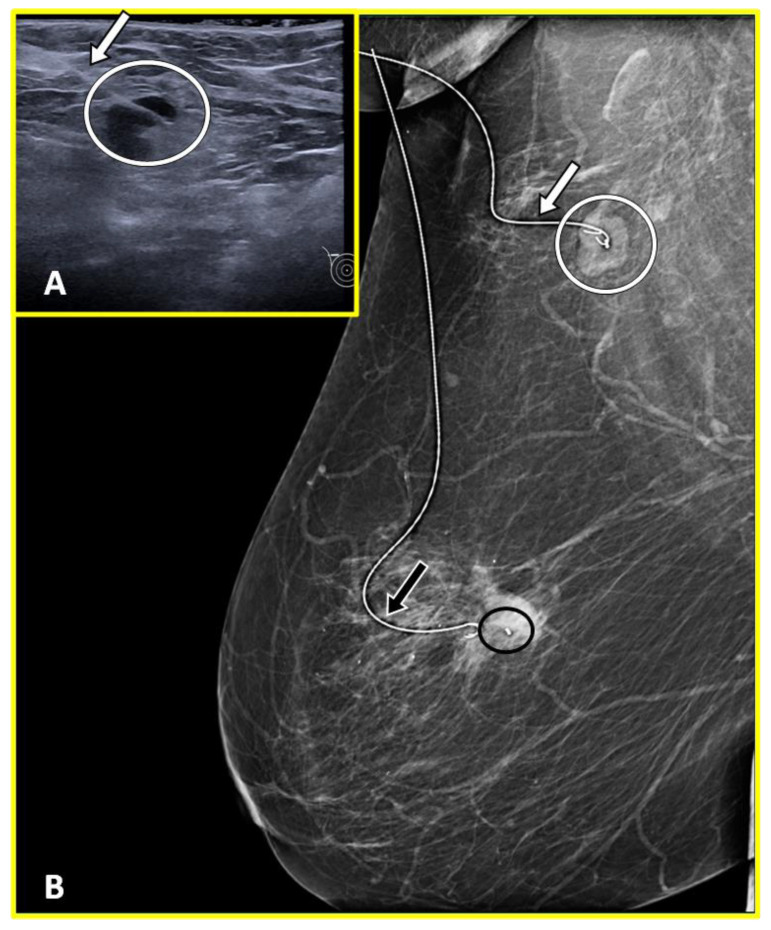
Preoperative wire localization in a 53-year-old woman with right breast tumor and axillary node metastasis. Ultrasound axial image (**A**) shows the metal wire which appears hyperechoic (white arrow) positioned inside the biopsy-proven pathological node (white circle). Post-procedural medio-lateral oblique (**B**) right mammogram confirmed the correct position of the wire (white arrow) within the pathological node (white circle). The wire position (black arrow) adjacent to the marker clip (black circle) within the tumor at 9 o’clock in the right breast is also shown.

**Figure 4 cancers-15-02130-f004:**
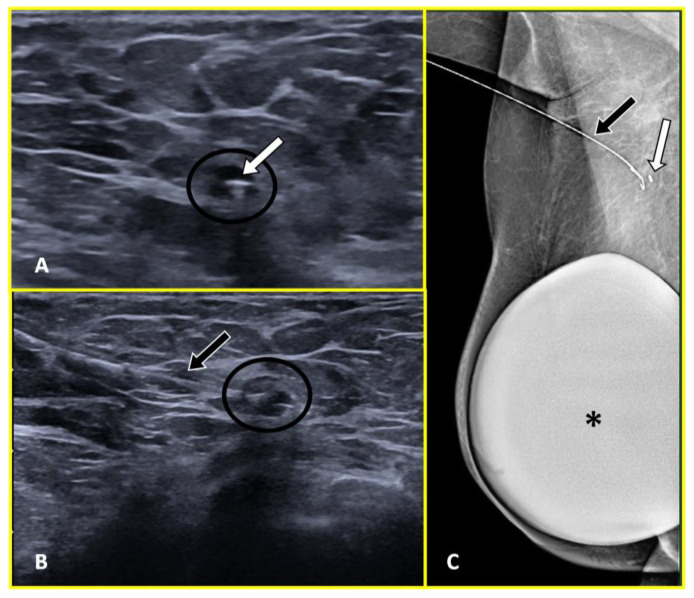
Preoperative wire localization in a 45-year-old woman with right axillary breast cancer recurrence. Ultrasound axial image (**A**) shows the previously positioned marker clip (white arrow) within the pathological node (black circle). Image (**B**) depicts the ultrasound-guided placement of metal wire (black arrow) inside the node (black circle). Post-procedural medio-lateral oblique (**C**) right mammogram demonstrated the correct position of the wire (black arrow) adjacent to the marker clip within the node (white arrow). A breast prosthesis is also visible (asterisk).

**Figure 5 cancers-15-02130-f005:**
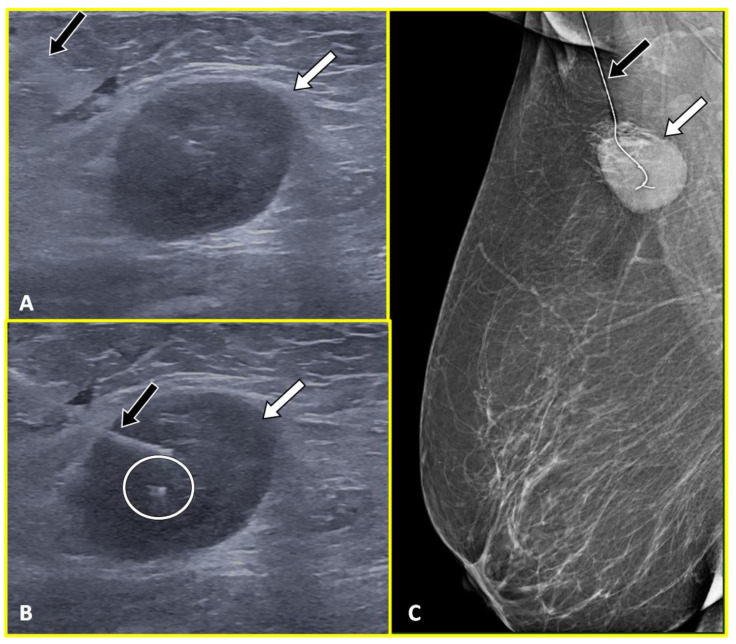
A 72-year-old woman with biopsy-proven right axillary node metastasis by a luminal B breast cancer for which the primitive tumor was not identified. Ultrasound preoperative wire localization (**A**,**B**) shows the wire (black arrow) positioned inside the pathological node (white arrow in (**B**)); a marker clip (white circle) within the node was previously placed. Post-procedural medio-lateral oblique (**C**) right mammogram confirmed the position of the wire (black arrow) within the node (white arrow).

**Figure 6 cancers-15-02130-f006:**
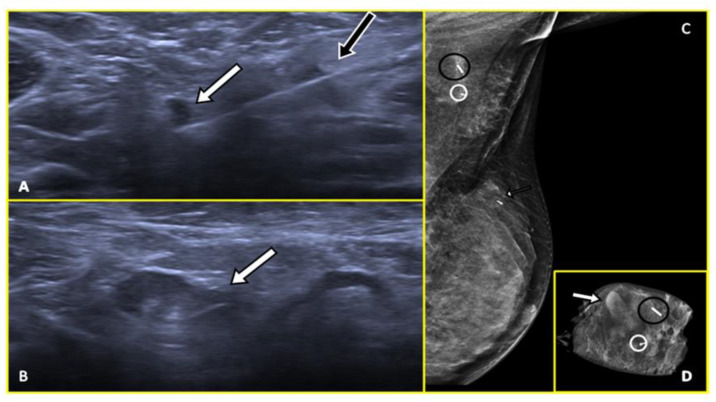
A 28-year-old patient with left breast cancer in the upper-outer quadrant and ipsilateral left axillary node metastasis treated with neoadjuvant chemotherapy. Ultrasound images (**A**,**B**) show the deployment of the magnetic seed in the pathological axillary node (white arrows) through an 18-gauge steel needle (black arrow). Subsequently, the medio-lateral oblique left mammogram (**C**) depicts the correct localization of the node with the magnetic seed (black circle). It is also visible the marker clip in the node (white circle) and the marker clip in the tumor of the upper-outer quadrant (white thin arrow), placed before the beginning of neoadjuvant treatment. The post-surgical specimen mammogram (**D**) confirmed the presence of the two markers (black and white circle) located in the pathological node (white thick arrow).

**Figure 7 cancers-15-02130-f007:**
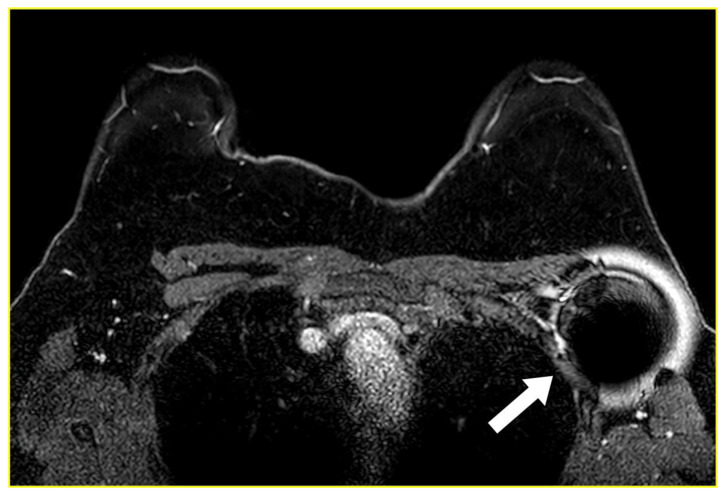
Gradient echo fat-suppressed contrast-enhanced T1-weighted axial image shows the ferromagnetic artifact (arrow) generated by a magnetic seed previously placed in a left axillary lymph node.

**Figure 8 cancers-15-02130-f008:**
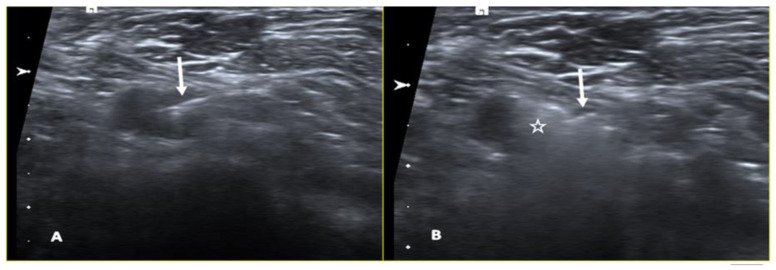
A 55-year-old patient with right breast cancer and ipsilateral axillary node metastasis treated with neoadjuvant treatment. The axial ultrasound image shows the injection of the radiotracer (^99m^Tc-Nanoscan) in the pathological node through a 21-gauge steel needle (arrow in (**A**,**B**)), which appears typically hyperechoic (star in (**B**)).

**Table 1 cancers-15-02130-t001:** False-negative rates (FNR) for SLNB in cases of conversion from positive LN at diagnosis to negative LN after NACT.

Prospective Trial	Overall FNR (%)	FNR Stratified by Number of SLNs (%)			FNR Stratified by SLN-Detection Technique (%)	
		1	2	3	Single Agent	Dual Agent
SENTINA	14.2	24.3	18.5	7.3	16	8.6
ACOSOG Z1071	12.6	31.5	21	9.1	20.3	10.8
SN FNAC	8.4	18.2	4.9 *	NR	16.0	5.2
GANEA 2	11.9 ^a^	19.4	7.8 *	NR	NR	11.9

^a^ A combined SLN detection method with blue dye and radiocolloid was recommended. * Reported as two or more. Abbreviations: NR: not reported; SLN: sentinel lymph node.

**Table 2 cancers-15-02130-t002:** Current recommendations of the main guidelines in patients with lymph node metastasis at the time of diagnosis down-staged to negative-node on clinical and ultrasound examination after neoadjuvant chemotherapy (NACT).

Guidelines	Staging Recommendations in Cases of Conversion from Positive LN at Diagnosis to Negative LN after NACT	Level of Evidence/ Grade of Evidence
**St. Gallen International Consensus Guidelines (SGICG)** [34]	TAD may avoid ALND if the TAD after NACT removes the marked node and one or two additional sentinel nodes, and all are negative. SLNB after NACT could be adequate only for patients with at least 3 or more negative SLNs.	Not provided
**American Society of Clinical Oncology (ASCO)** [35]	SLNB is recommended to restage the axilla. Restaging can be achieved by placing a biopsy clip into the biopsied positive node at diagnosis and localizing it at surgery along with sentinel node biopsy or, in institutions where the use of biopsy clips for nodes is not available, by performing sentinel node biopsy with a dual tracer and excising at least three sentinel nodes.	Evidence quality: low; Strength of recommendation: weak
**National Comprehensive Cancer Network (NCCN)** [13]	Panel recommends pathologic confirmation of malignancy using ultrasound-guided fine-needle aspiration or core biopsy of suspicious nodes with clip placement. These patients may undergo SLNB with the removal of the clipped lymph node. A relatively high false-negative rate (FNR) (>10%) can be improved by marking biopsied lymph nodes to document their removal, using a dual tracer, and by removing 3 sentinel nodes (targeted ALND). When sentinel nodes are not successfully identified, the panel recommends level I and II axillary dissections be performed for axillary staging.	2B
**Breast Committee of the German Gynaecological Oncology Working Group (AGO)** [36]	Suspicious lymph nodes should be evaluated before NACT by core needle biopsy and marker placement. SLNB only may be performed only in individual cases (AGO+/−); however, if 3 or more negative SLNs alone were removed and nodal radiotherapy was performed, the local recurrence rate is very low. ALND can be performed (AGO+) but may be harmful. TAD can be performed (AGO+). However, in case of extensive axillary tumor load (≥4 suspicious nodes) at presentation it should be used with caution (AGO+/−).	2B
**American Society of Breast Surgeons (ASBS)** [37]	SLNB is suitable. The false-negative rate of SLNB is minimized by the retrieval of >2 SLN, by dual mapping, and by retrieval of the biopsied/clipped node. ALND is indicated for patients who are cN0 but SLN+.	Not provided

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
