# Peer review of "Image-Guided Localization Techniques for Metastatic Axillary Lymph Nodes in Breast Cancer; What Radiologists Should Know"

_cancers, 2023, doi:10.3390/cancers15072130_

Round 1

Reviewer 1 Report

"Unfortunately, the equipment to perform LNs localization, especially the probe and console, are burdened by high costs." I don't agree with this statement, if there is a Nuclear Medicine department, certainly the sentinel lymph node research technique is performed with a radioactive tracer and therefore there is a device in the department equipped with probes and console. However, the cost is not very high and can be completely recouped with radio-guided surgery procedures, not exclusively for the breast.

"Despite a very low risk, emergency treatment with iodine is required in case of semen rupture to avoid damage to the thyroid gland" This is only true in theory, since the amount of radioactivity contained in the seed is absolutely negligible as a potential risk for the thyroid

Author Response

I want to thank the reviewer for improving the quality of the paper through his suggestions.

"Unfortunately, the equipment to perform LNs localization, especially the probe and console, are burdened by high costs." I don't agree with this statement, if there is a Nuclear Medicine department, certainly the sentinel lymph node research technique is performed with a radioactive tracer and therefore there is a device in the department equipped with probes and console. However, the cost is not very high and can be completely recouped with radio-guided surgery procedures, not exclusively for the breast.

This is a correct observation; we add this in the text (pag 11).

"Despite a very low risk, emergency treatment with iodine is required in case of semen rupture to avoid damage to the thyroid gland" This is only true in theory, since the amount of radioactivity contained in the seed is absolutely negligible as a potential risk for the thyroid.

He have specified the very low risk in the text (pag 11).

Reviewer 2 Report

I would like to thank the authors for writing a detailed review of axillary lymph node localization methods after NACT. The authors have very thoroughly discussed the historical perspective and the current state of research on the topic. The role of axillary lymph node metastasis has long been the subject of controversy in the literature. There are several strategies for marking, but to date, none has proven to be better, also on TAD there are many perspectives. I appreciate the paper.

I have some suggestions to address to make the manuscript Interesting for all breast physicians:

- Expand the introductory part on axillary cord management and its classifications, not all breast surgeons have availability of these methods, in centers with low operative volumes.

- The role of vital indocyanine green dye and Superparamagnetic Iron Oxide is not considered.

Therefore, I recommend introducing these manuscripts:

Pelc Z, Skórzewska M, Kurylcio M, Nowikiewicz T, Mlak R, Sędłak K, Gęca K, Rawicz-Pruszyński K, Zegarski W, Polkowski WP, Kurylcio A. A Propensity Score Matched Analysis of Superparamagnetic Iron Oxide versus Radioisotope Sentinel Node Biopsy in Breast Cancer Patients after Neoadjuvant Chemotherapy. Cancers (Basel). 2022 Jan 28;14(3):676.

Cirocchi et al. New classifications of axillary lymph nodes and their anatomical-clinical correlations in breast surgery. World J Surg Oncol. 2021 Mar 29;19(1):93. doi: 10.1186/s12957-021-02209-2. 

Staubach P, Scharl A, Ignatov A, Ortmann O, Inwald EC, Hildebrandt T, Gerken M, Klinkhammer-Schalke M, Scharl S, Papathemelis T. Sentinel lymph node detection employing indocyanine green using the Karl Storz VITOM®️ fluorescence camera: a comparison between primary sentinel lymph node biopsy versus sentinel lymph node biopsy after neoadjuvant chemotherapy. J Cancer Res Clin Oncol. 2021 Jun;147(6):1813-1823. doi: 10.1007/s00432-020-03461-x.

2.            Both the objectives, methods, and conclusions of this research are broad and radical.

3.            The figures are of good quality and standardized.

I consider it interesting after this review

Author Response

I want to thank the reviewer for improving the quality of the paper through his suggestions.

- Expand the introductory part on axillary cord management and its classifications, not all breast surgeons have availability of these methods, in centers with low operative volumes.

Added in the text (pag 1)

- The role of vital indocyanine green dye and Superparamagnetic Iron Oxide is not considered.

Added in the text (pag 1 and 2)

Therefore, I recommend introducing these manuscripts:

Pelc Z, Skórzewska M, Kurylcio M, Nowikiewicz T, Mlak R, Sędłak K, Gęca K, Rawicz-Pruszyński K, Zegarski W, Polkowski WP, Kurylcio A. A Propensity Score Matched Analysis of Superparamagnetic Iron Oxide versus Radioisotope Sentinel Node Biopsy in Breast Cancer Patients after Neoadjuvant Chemotherapy. Cancers (Basel). 2022 Jan 28;14(3):676.

Cirocchi et al. New classifications of axillary lymph nodes and their anatomical-clinical correlations in breast surgery. World J Surg Oncol. 2021 Mar 29;19(1):93. doi: 10.1186/s12957-021-02209-2. 

Staubach P, Scharl A, Ignatov A, Ortmann O, Inwald EC, Hildebrandt T, Gerken M, Klinkhammer-Schalke M, Scharl S, Papathemelis T. Sentinel lymph node detection employing indocyanine green using the Karl Storz VITOM®️ fluorescence camera: a comparison between primary sentinel lymph node biopsy versus sentinel lymph node biopsy after neoadjuvant chemotherapy. J Cancer Res Clin Oncol. 2021 Jun;147(6):1813-1823. doi: 10.1007/s00432-020-03461-x.

The references were introduced as suggested (pag 1 and 2)